# Why does it take so long for rare disease patients to get an accurate diagnosis?—A qualitative investigation of patient experiences of hereditary angioedema

**Moeko Isono[1], Minori Kokado[1,2], Kazuto Kato****[1]\***

**1** Department of Biomedical Ethics and Public Policy, Graduate School of Medicine, Osaka University, Suita, Osaka, Japan, **2** Faculty of Pharmacy, Kobe Pharmaceutical University, Kobe, Hyogo, Japan

\* kato@eth.med.osaka-u.ac.jp

## Abstract

### Introduction

Many patients with rare diseases experience a diagnostic delay. Although several quantitative studies have been reported, few studies have used a qualitative approach to directly examine how patients with rare disease obtain a diagnosis and why it takes many years. In this study, we focused on hereditary angioedema (HAE), which has been reported to have long diagnostic delays, despite the knowledge that not having an accurate diagnosis can cause life-threatening problems.

### Objective

The objective of this study was to analyze patients' experiences and elucidate why it takes a long time to reach a diagnosis of HAE. We also aimed to propose possible solutions for the problem.

### Methods

A qualitative study using semi-structured interviews was conducted. Nine patients who took over 5 years from the presentation of initial symptoms to an HAE diagnosis participated. The contents of the interviews were subjected to an inductive contents analysis.

### Results

By analyzing the patients' struggles that were experienced during the undiagnosed period, three themes were generated: (1) acceptance and resignation towards their conditions, (2) proactive search for a cause, and (3) independent efforts outside of the hospital. While a few patients continued to seek out a diagnosis during the undiagnosed period, many had become accustomed to their health condition without suspecting a rare disease.

**Data Availability Statement:** Data are not publicly available due to concerns of participant confidentiality. To allow public access to the data is in contradiction to the ethical agreement signed by

participants and approved by the Osaka University Clinical Research Review Committee. Some of the aggregated data may be available with an approval of the Review Committee. The Osaka University Clinical Research Review Committee can be contacted at: rinri@hp-crc.med.osaka-u.ac.jp.

**Funding:** KK: JP17K19812 Japan Society for the Promotion of Science Grants-in-Aid for Scientific Research (KAKENHI) https://www.jsps.go.jp/. The funders had no role in study design, data collection and analysis, decision to publish, or preparation of the manuscript.

**Competing interests:** The authors have declared that no competing interests exist.

## Conclusions

We found that one of the most important factors related to the prolonged undiagnosed period is the lack of suspicion of a rare disease by patients and their medical professionals. While current policies tend to focus on the period from suspecting rare diseases to the time of a clear diagnosis, our results strongly suggest that measures are needed to facilitate patients and clinicians to become aware of rare diseases.

## Introduction

Many patients with rare diseases (RDs) worldwide are struggling to find a diagnosis. Currently, there are approximately 10,000 types of RDs, and 473 million people are affected by RDs worldwide [1]. For these RD patients, one of the most pertinent issues is the time required to reach a correct diagnosis. RD patients can be divided into two groups: (1) 'not yet diagnosed' refers to patients with an undiagnosed status that should be diagnosed for known diseases but have not been because the patients have not been referred to the appropriate clinician; and (2) 'undiagnosed' refers to patients for whom diagnostic tests are not yet available since the diseases have not been characterized and the cause(s) has not been identified [2]. Even if a patient belongs to group (1), it takes an average of 4–9 years to reach a correct diagnosis [3–6]. In low- and middle-income countries, where resources and specialized services are known to be very limited, the undiagnosed period is estimated to be probably even longer [7].

While many quantitative studies have been reported, comparatively fewer qualitative studies were conducted regarding the undiagnosed period of patients. Several patient organizations have conducted questionnaire surveys and reported cross-disease data [4–6, 8]. Moreover, many specialists and patient groups have conducted disease-specific quantitative surveys [9]. These quantitative surveys showed the means or medians of time to diagnosis and a list of misdiagnoses. A few other studies describe stories of diagnostic delay of individual patients by a qualitative approach [5, 10–12]. Thus, it is difficult to concretely grasp the commonality of diagnostic delay.

Recently, national projects have been launched in several countries based on genetic analysis, using next-generation sequencing (NGS), to shorten the diagnostic delays for RDs. These include the Undiagnosed Diseases Program/Network (UDP/UDN) in the United States [13, 14], the Finding of Rare Disease Genes (FORGE) program in Canada [15] and Deciphering Developmental Disorders (DDD) in the United Kingdom [16]. These are projects driven by national policy, whereas the North-West University's Centre for Human Metabolomics (CHM) of South Africa is in the process of establishing the first RD biobank that will facilitate early diagnosis [7].

In Japan, the Initiative on Rare and Undiagnosed Disease (IRUD) was launched in 2015 [17]. Additionally, the Ministry of Health, Labour, and Welfare issued a notification in 2018 to achieve early diagnosis of rare and intractable diseases. This notification requests every prefectural government to develop its own medical care delivery system. The medical care delivery system must connect patients whose conditions are difficult to diagnose by standard medical practices to a higher-order medical facility and national networks, including IRUD [18].

In this context, the following question has arisen: Are the government efforts enough to solve the problem of patients being undiagnosed? We believe that experiences of the patients before they reach a correct diagnosis are not adequately understood. Given the expected

diversity and complexity of the experience, it is likely that they are not grasped by medical professionals, policy makers, and other stakeholders. Therefore, it was expected that we could obtain insights into workable solutions for this problem by deepening our understanding of the experiences of individual patients and elucidating common features among them. Thus, we studied patients' paths to diagnosis, why the diagnosis took so many years, and whether the current measures are appropriate.

Among the RDs, we decided to focus on hereditary angioedema (HAE) and tried to understand the patients' experience of being undiagnosed. There were three reasons for this: (1) the average length (years) of the undiagnosed period in HAE is longer than in many other RDs; (2) HAE symptoms are diverse; and (3) not having diagnosis of HAE is a life-threatening issue (see below). We believed that reasons (1) and (2) could justify the diversity of HAE patients' experiences during the undiagnosed period. Consequently, our results may help clarify some of the common features, experienced by other RD patients during the undiagnosed period. Furthermore, bearing in mind reason (3), we considered that HAE needs to be adressed most urgently among all RDs.

HAE is a rare, potentially life-threatening genetic condition. It can be categorized into 3 different types, including HAE with deficit C1-inhibitor (C1-INH) levels (type 1), HAE with dysfunctional C1-INH (type 2), and HAE with normal C1-INH function (HAE-nC1-INH). Most patients belong to type 1 or 2, but within these two categories 85% are type 1. HAE-nC1-INH is very rare and not reported enough.

Type 1 and type 2 are autosomal dominant conditions (although approximately 25% of patients have no history). The estimated combined prevalence is approximately 1:50000, with no reported differences among different ethnic groups [19]. The percentage of female patients has been reported to range from 55% to 69% [20–23]. HAE needs to be differentiated from acquired angioedema (AAE), which is suspected if there is no family history of the disease and if the onset is after the age of 30 years [19].

Clinically, all forms of HAE are characterized by recurrent episodes of swelling or edema that target different regions of the body. The most commonly involved organs include the skin, upper respiratory tract, oropharynx, and gastrointestinal tract. Severe edema in the airways may become life-threatening. Acute attacks may be treated by a bradykinin B2 receptor antagonist, icatibant, and/or plasma-derived C1-INH concentrate.

Early diagnosis of HAE is essential. Mortality is estimated to be three times higher in patients who are undiagnosed than in those who are diagnosed [24]. In addition, at least two recent fatal cases of undiagnosed patients have been reported in Japan [25, 26]. However, it was reported that the average undiagnosed period among HAE patients in Japan was 13.8 years as per 2014 survey [20] and 15.6 years as per 2020 survey [27].

The objective of this study was to grasp and analyze HAE patients' experiences, focusing on two points: (1) what actions they took to relieve or ameliorate their symptoms and what medical care they received during the undiagnosed period, and (2) how their current diagnosis was reached. Based on these results, we also aimed to understand why there is such a protracted period before an accurate diagnosis of HAE is reached. Then, we discuss the factors contributing to the prolonged undiagnosed period. The findings of this study will help us understand why patients with RDs experience such a long delay before they are accurately diagnosed and find solutions to shorten the undiagnosed period.

## Material and methods

To understand experiences of HAE patients during their undiagnosed period, a qualitative research approach using semi-structured interviews was employed.

## Recruitment

Patients were eligible to participate if they were 20 years or older and met the following two criteria: (1) patients who were diagnosed with HAE type 1 or 2 (characterized by C1-INH deficiency/dysfunction); and (2) >5 years had elapsed from the initial appearance of symptoms to an HAE diagnosis. Patients who met the above criteria were invited to participate in this study from April 2019 to October 2019. We advertised the study (information dissemination) in the following three ways: (1) notice posted on the online rare disease research platform RUDY JAPAN [28], (2) notice distributed via e-mail from the HAE patient organizations, and (3) notice given to patients from HAE specialists. In addition to information dissemination, we conducted snowball sampling. A 3,000-yen (approximately 29 USD) voucher was offered as compensation to all the participants (two of them declined).

Since the purpose of this study was to understand the phenomenon, and not to construct a theory, we did not aim for data saturation. By 6 months after the beginning of information dissemination, we obtained nine participants and stopped recruitment. This is because we had already used all the recruitment methods possible and had sent out information to the same group several times. We considered that most patients who wanted to voluntarily participate had already contacted us and sending out more information could hinder benefit of patients who were not willing to participate in this study.

## Data collection

An interview guide was developed based on the aim of this study. The main topics covered were patients' experiences from their initial symptoms until diagnosis (especially whether they had visited hospitals or not; if yes, the diagnosis/explanation and treatment at the hospital) and changes in their perceptions about their physical problems.

One-to-one semi-structured interviews were conducted for approximately 90 minutes. Interviews were conducted either face-to-face or using a web conferencing system (Zoom). All interviews were audio-recorded and transcribed verbatim with the permission of the participants.

## Data management and analysis

We analyzed the data using content analysis [29]. The approach used was mainly inductive and manifest analysis, involving coding and grouping excerpts, describing the same phenomena to produce descriptive summaries of the data. However, given the purpose of the study, in a relatively early stage, we separated the data into two major categories: "patient struggles during the undiagnosed period," and "how they were able to reach a diagnosis of HAE." Subsequently, all analyses were conducted inductively. Coding and initial grouping into categories were performed by one researcher. Two researchers with experience in qualitative research, ensured that the analysis was appropriate for the objective of the study.

In this content analysis, we counted and described the frequencies of statements of each theme and sub-theme. This transparent reporting allows us to present to the readers with substantial evidence for each of the descriptive summaries. We intended to make our reporting more precise, rigorous, and scientific by describing the frequency of statements [30].

## Ethical approval and consent to participate

Approval to conduct this research was granted by the Osaka University Clinical Research Review Committee (18534–5). Written informed consent for research participation was obtained from all interviewees.

## Results

### Characteristics of the participants

Nine patients participated in this study (hereafter, we call them participants). The average length of their undiagnosed period was approximately 23 years. As shown in Table 1, most participants (8/9) visited a hospital within a year from the initial appearance of symptoms. However, it took many years for the diagnosis of HAE from their first visit to the hospital. Most participants (8/9) visited the hospital repeatedly without knowledge of HAE. The only exception was Participant F; he was aware of HAE before initial symptoms appeared, but had a hard time before getting an official diagnosis of HAE (see below for further description).

### "Past HAE symptoms" described in the interview

The symptoms experienced by the nine participants in the undiagnosed period were diverse (Table 1). During the interviews patients were asked to retrospectively discuss the symptoms they experienced during this undiagnosed period. Many remarked that they were not always sure, in the case of abdominal attacks, whether they were caused by HAE or something else. Particularly, they lacked confidence to clearly suggest "past symptoms" as definitively caused by HAE when there was no visible swelling. Since this study aims to understand patients' experience, we describe "experience of HAE symptoms" based on participants' perspectives.

Consequently, many participants experienced visible swelling and abdominal symptoms (9/9 and 8/9, respectively). Almost all participants clearly talked about visible swelling as a symptom of HAE. However, they were not confident when they talked about their abdominal symptoms. They made comment such as, "I don't know if it was due to HAE or not," explaining that they could not always confidently distinguish HAE from other possibilities such as Cyclic

**Table 1. Participant characteristics.**

| Participant | Generation | Sex | The time between the initial symptoms and the first visit to the hospital (years) | The time between the first visit to the hospital and the diagnosis of HAE (years) | Year when he/she got the diagnosis of HAE | HAE symptoms that patients experienced in the undiagnosed period | | | |
|---|---|---|---|---|---|---|---|---|---|
| | | | | | | Visible swelling[*1] | Abdominal symptoms[*2] | Laryngeal swelling[*3] | Other symptoms |
| A | 50s | Female | Less than 1 | 39–40 | 2012 | + | ++ | ++ | |
| B | 30s | Female | Less than 1 | 13 | 2008 | + | ++ | + | |
| C | 50s | Female | Less than 1 | 26 | 1994 | ++ | ++ | − | Joint pain+ |
| D | 50s | Female | Less than 1 | 20 | 2017 | + | − | − | |
| E | 40s | Female | Less than 1 | 20 | 2012 | ++ | ++ | + | |
| F | 60s | Male | 31–32 | 1–2 | 2015 | + | + | − | Nasal congestion++ |
| G | 20s | Female | Less than 1 | 8 | 2016 | + | + | − | |
| H | 30s | Female | Less than 1 | 17 | 2003 | ++ | + | ++ | |
| I | 50s | Female | Less than 1 | 28 | 2017 | + | ++ | − | |

+ Indicates the topics mentioned by the participants

++ Indicates the topics that specifically highlight the severity of their symptoms; − Indicates the topics not mentioned by the participants.

[*1] "Visible swelling" includes swelling of the extremities, finger, wrist, feet, and face, among others.

[*2] "Abdominal symptoms" include abdominal pain, nausea/vomiting, and constipation.

[*3] "Laryngeal swelling" includes dyspnea/breathlessness.

HAE, hereditary angioedema.

Vomiting Syndrome, stomach flu, or menstrual pain. Moreover, several participants reported experiencing a variety of health problems as a child, including frequent colds or getting a positive urine test during a routine check-up. In this context, they found it difficult to distinguish between what might have been symptoms of an RD and those related to being an "unhealthy child". Regarding these episodes, one of them added a comment that "the current doctor said that these past problems and episodes would not be related to HAE, but I am not certain about it."

*"Aside from the limbs, I finally understand the other symptoms [of HAE]–namely, the vomiting, diarrhea, and nausea. It became clear that they were part of my attack this year. The reason I understood this was because I tried treating them with FIRAZYR®\* every time [any problems occurred]. I finally understood that this was an attack!" (Participant I)*

\* FIRAZYR® is a brand name of icatibant, bradykinin B2 receptor antagonist to be injected, licensed for acute attack treatment.

## Alternative diagnoses

Before the final diagnosis of HAE, participants were given various alternative diagnoses to explain their symptoms (Table 2). Visible swelling of several participants had been diagnosed as an allergy. Abdominal symptoms were diagnosed with various diseases or symptoms, such as stomach flu or appendicitis. Laryngeal symptoms were attributed to asthma and colds. Several participants had received alternative diagnoses, while others did not have clear diagnoses or explanations.

## Analysis of participants' experiences (A) — struggles during the undiagnosed period

In the following sections, we present results of the analysis of participants' experiences. It consists of two sections: (A) their struggles during the undiagnosed period and (B) how they were able to reach a diagnosis of HAE.

During their long undiagnosed period, patients suffered from symptoms and faced various challenges. We describe these experiences according to three themes: (A-1) acceptance and resignation towards their condition, (A-2) proactive search for a cause, and (A-3) independent efforts outside of the hospital. Table 3 shows the subthemes contained within each theme.

**Table 2. Alternative diagnoses in the undiagnosed period.**

| Visible swelling | Abdominal symptoms | Laryngeal swelling |
|---|---|---|
| • Insect bite | • Food poisoning | • Asthma |
| • Allergy (egg, dust, or tick) | • Irritable bowel syndrome (IBS) | |
| • Erythema annulare centrifugum | • Endometriosis | |
| | • Cyclic Vomiting Syndrome | |
| | • Appendicitis | |
| | • Gastric and duodenal ulcers | |
| | • Intestinal obstruction (adhesion of the surgical scars of appendicitis) | • Cold |
| | • Constipation | |
| | • Stomach flu | |
| | • Acute abdomen | |

**Table 3. Themes and sub-themes of participant experiences (A).**

| Themes | Sub-themes |
|---|---|
| **(A-1) Acceptance and resignation towards their condition** | *[Visible swelling]* Getting used to lack of explanation or diagnosis |
| | *[Abdominal symptoms]* Repetition of their usual coping methods |
| **(A-2) Proactive search for a cause** | Proactive search for a cause by a patient |
| | Suggestion to search for a cause by a doctor |
| **(A-3) Independent efforts outside of the hospital** | – |

**(A-1) Acceptance and resignation towards their conditions.** In this theme, participants' experiences were divided into two sub-themes depending on the site and manifestation of the symptoms. One was visible swelling, and the other was abdominal symptoms. It was revealed that participants went through various experiences and thoughts, and they finally accepted their conditions that was, in a way, characteristic of each symptom (Table 3).

For visible swelling, seven participants visited hospitals (e.g., department of dermatology). Five participants described the motivation as being "because I hadn't the slightest idea what the reason was" or "because people around me recommended it." However, only little or no clear diagnosis or explanation was provided in the hospital, without being referred to a higher-level hospital. Only one participant was given another diagnosis. Two episodes of referral to more specialized hospitals were reported, but neither of them resulted in a correct diagnosis. Regardless of the diagnosis, participants were left with questions about the cause of their symptoms and visited several different hospitals.

> *"When I was 16- and 17-year-old, I had swelling in my limbs about once a year. I didn't know what caused it but I decided to go to a dermatologist. The doctor said it might be an allergic reaction. They often said "maybe", or "I don't really know why". They said, "It might be an insect bite, or an allergic reaction, or something like that." I didn't get any clearer diagnosis than that and I didn't receive any medication." (Participant E)*

After several visits to hospitals, five participants stopped visiting the hospital. The reasons for this were that there was nothing unusual in the test results, which led doctors and participants to think that it was not a serious problem; they became accustomed to the fact that they could not get a clear diagnosis and medication when they went to the hospital; they learned from their experiences that this swelling would disappear without treatment after several days.

Although they did not receive any clear diagnosis, some participants thought that symptoms were due to something more. For example, Participant H considered her symptoms to be abnormal and thought it was a "weird disease" in her heart. However, she became accustomed to not being diagnosed in a neighboring hospital and stopped seeing a doctor. She said, "I had given up and put up with having to deal with this weak constitution for the rest of my life."

> *"I always thought that I had an allergic constitution which I have to live with throughout my life. [. . .]. Since no medical department that I had found could give me an answer, I started to endure the swelling at home. [. . .]. No abnormality in general blood tests relieved me and made me think that it could not be a serious disease." (Participant H)*

For abdominal symptoms, seven participants visited hospitals (mainly the general medicine departments). Six mentioned their motivation as "wanting to do something about this pain,"

that is, indicating that they were mainly seeking treatment. Many participants emphasized very severe pain and suffering, such as, "I collapsed," and "I felt like I was dying." Two of them described multiple visits to the emergency room, and one participant had undergone an abdominal operation.

Regarding abdominal symptoms, two participants (Participant A and E) underwent treatments for diagnoses other than HAE administered by their doctors. For example, Participant E was always admitted to the same hospital when she had severe abdominal symptoms. She was treated for the diagnosis of gastric and duodenal ulcers, but the medication did not work. The doctor expressed doubts about the treatment and, after several tests, changed the medication; even then, it did not work. Participant E also wondered why the treatment did not work but believed in the diagnosis because of the doctor's diligence.

Four participants had never been diagnosed by doctors with a specific disease name. They were provided explanations, such as symptoms being "stress-related" or the result of the "stomach flu," and prescribed palliative drugs for the symptoms. A few participants actively sought a diagnosis and treatment, while the others did not. However, participants with severe symptoms were repeatedly hospitalized for several days when a symptom occurred, regardless of their activity for seeking proper diagnosis.

*"One of the doctors saw me and said, "You should be hospitalized," and when I got better, they said, "You can leave the hospital". When the symptoms re-appeared, they said, "You should be hospitalized". I didn't have an attending doctor, and the doctors would say, "Hmm, I wonder why this is happening" and so on. [. . .]. "It happened again!" I really blamed myself, psychologically. [. . .]. I thought that it was happening because I had a weak constitution, or that I was stressed. . . basically that I lacked self-control." (Participant B)*

Two participants described that they stopped visiting hospital, despite symptoms such as severe abdominal pain. These participants were affected by what a doctor or people around them explained as, "It's a psychosomatic disorder."

*"My friend told me that "If they didn't find anything during the check-up, the vomiting must be stress-related." I thought they might be right, and I didn't visit the hospitals so frequently." (Participant I)*

As mentioned above, Participant I was told that it was a psychiatric problem by her friend after a test at the hospital, while Participant A was repeatedly told that it was weak constitution or psychiatric problem by doctors after medical consultations only; tests were not conducted in detail. After that, Participant A experienced a detailed checkup and received a diagnosis of irritable bowel syndrome (IBS) and took a drug for IBS. However, the drug made her health condition worse. After this episode, Participant A gave up on visiting hospitals, thinking that it would not help, and came to endure the symptoms at home for about 20 years. Participant A finally revisited a hospital after vomiting for 36 hours and falling into a state of shock.

**(A-2) Proactive search for a cause.** While many participants came to accept their strange health condition and were accustomed to coping with it, a few participants and their doctors proactively searched for a disease name.

Three participants suspected that their physical problems might have occurred due to an underlying and unrecognized cause. Two of them had been actively visiting hospitals to look for a correct diagnosis. Participant C had visited all the large hospitals in the neighboring regions, searching for a diagnosis for two reasons: (1) the doctors questioned and were concerned about her condition upon seeing her visible swelling and abdominal pain, but none of

them could give her a diagnosis, and (2) she recognized that her condition clearly differed from that of other people. Participant I underwent regular gastroscopy in search of the cause. Moreover, when she visited the doctor with concerns about another health issue (high risk of blood clots), she asked the doctor about a connection between it and her visible swelling. She sought an underlying and unrecognized cause for various reasons. One was that she suspected a possibility of genetic disorder because her blood relatives had similar abdominal problems. She was also concerned about her many physical problems, such as visible swelling, abdominal symptoms, back pain, and fatigue.

Two participants (Participant A and G) were advised to undergo tests and were admitted to hospitals because their doctors thought they had unexplained symptoms. However, this suggestion was not received well. One reason was that participants were reluctant to be hospitalized. They felt that it was costly and time-consuming, considering that the doctor did not have a specific disease in mind and that the symptoms had already subsided. For this reason, Participant G did not undergo hospitalization for tests. In contrast, despite not wanting to be hospitalized, Participant A underwent multiple hospitalizations for tests that took an average of 10 days. A serious disease was suspected, and she underwent various tests, repeatedly, most of which were quite invasive. However, the doctors did not come up with HAE as a possibility. She lost hope in medicine and thought that it was just time-consuming. Thus, she chose not to visit the hospital unless the symptoms were unbearable, and she felt that she would go into shock again. Despite this choice, she was still hospitalized approximately twice a year for 5 years, prior to being diagnosed.

**(A-3) Independent efforts outside of the hospital.** Participants whose conditions were not improved by medical treatments attempted to cope for themselves outside hospital. Their medical professionals in those days did not know their behaviors.

One of the most frequent behaviors was trying to understand the triggers of the symptoms. Three participants attempted to determine the relationship between their diet or lifestyle and the occurrence of symptoms. The most severe example was the case of Participant A. She avoided a particular ingredient (e.g., wheat) to control her health condition. However, this effort resulted in severe weight loss so much so that she developed amenorrhea and anemia.

Another behavior was seeking out information or people having similar problems. Three participants did this. Those include examples of searching for similar patients via a paid site on the Internet and of asking for any idea regarding the symptoms by showing a picture of the swelling to her friends.

## Analysis of participants' experiences (B) — how they were able to reach a diagnosis of HAE

As described above, participants experienced various struggles including repeated visits to hospitals, endurance of symptoms at home, and performing activities to cope with problems without visiting hospitals. Finally, HAE diagnosis was reached in various ways.

**(B-1) Obtaining a diagnosis just after knowing about HAE.** Seven participants suffered from their health condition for a long time, because they were unaware about HAE. Contrastingly, despite knowing about HAE, two participants experienced a period in which they did not receive an HAE diagnosis (see the further description below). Table 4 lists how participants were able to reach a diagnosis of HAE.

Participants generally expressed positive feelings about receiving a diagnosis of HAE. First, they mostly felt delighted and relieved that a cause was found and that a treatment was available. Second, they felt concerns about their future life and their family (especially their children). They also experienced disappointment that they had not been given a correct diagnosis

**Table 4. How participants reached a diagnosis of HAE.**

| | |
|---|---|
| Patient's effort to seek a diagnosis paid off | *[Visible swelling]* The patient finally found a doctor who knew about HAE by continuing visits to larger hospitals. |
| | *[Visible swelling]* A friend who knew of the symptoms of the participant found an advertisement of HAE and mentioned it to the participant. |
| The diagnosis was made as an indirect result of seeking a way to deal with symptoms | *[Abdominal symptoms]* During emergency transport, a patient showed a doctor a picture of a swollen limb and asked if it is related. |
| | *[Abdominal symptoms]* Another HAE patient who was diagnosed before in the other area relocated and saw the hospital where the participant often visited. |
| | *[Abdominal symptoms and laryngeal swelling]* The emergency department that the patient used to visit was notified of the HAE awareness campaign. |
| | *[Laryngeal swelling]* At the hospital where the patient was brought for emergency treatment, doctors from all departments came together to examine the patient and made a diagnosis. |
| The patients were aware of the existence of HAE from a blood relative | Blood relatives were suspected or diagnosed with HAE (2 cases). |
| The patient knew about HAE by himself | Participant made a diagnosis for himself after when symptoms became typical for HAE. |

*[]* indicates which symptoms specifically contributed to the episode that led to reaching an HAE diagnosis. HAE, hereditary angioedema.

sooner. However, one participant said, "I am glad to have a correct diagnosis, but the diagnosis also increased my anxiety about the possibility of life-threatening HAE attacks. For me, it was probably a good thing to reach the diagnosis in that the treatment was recently established."

**(B-2) The undiagnosed periods after knowing about HAE.** As special cases, two participants had learned of HAE but did not receive a diagnosis on their own soon. The experience of Participant F, an extreme exception among all participants, is described below.

In the case of Participant F, he learned the name of the disease before the onset of his own HAE symptoms, but it took approximately 35 years for him to be diagnosed. Participant F learned about HAE in a lecture when he was a student. At that time, he suspected that his father's symptoms were caused by HAE and asked the professor a question; however, the professor denied the possibility that he had the disease, saying that HAE only affected women, which is not the case for HAE type 1 and 2. Later, Participant F himself became troubled by the symptoms of nasal congestion to the extent that he considered surgery when he was in his 50s, at which time, due to his profession, he saw a pharmaceutical company's campaign to raise awareness about HAE. After that, he actively suspected that he had HAE and asked his doctor to test him for HAE before being prepped for surgery; however, because his symptoms were mainly nasal congestion, which is atypical for HAE, the doctor did not understand the urgency and importance of the test and did not proceed to perform the actual test. He went to several hospitals and departments to seek a doctor who understood HAE. He then experienced swelling in his face (this was his first swelling on the face), showed pictures of them to his doctor, and eventually received a diagnosis for HAE.

## Discussion

### Principal findings

This study revealed the details of experiences of patients with HAE during the undiagnosed period from various perspectives, such as their encounters with medical care and psychological

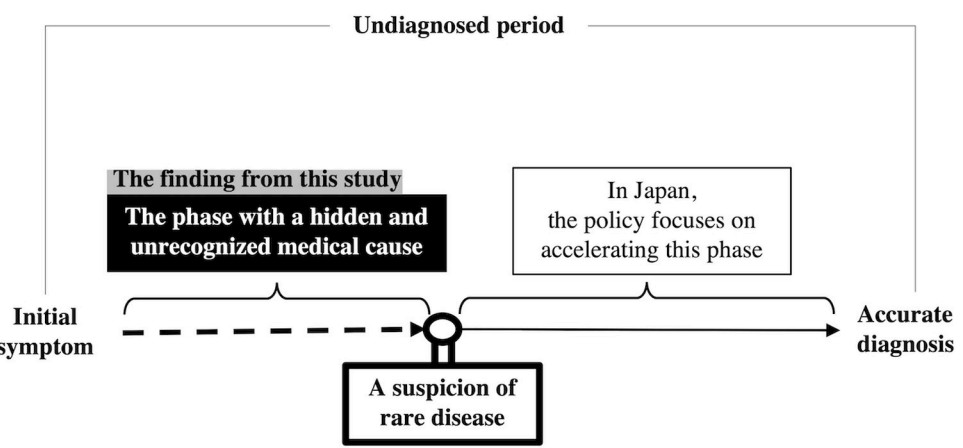

**Fig 1. Diagram illustrating problems leading to delayed diagnosis of rare diseases.**

conflicts. As a result, we identified two major factors that contributed to prolonged undiagnosed periods: (1) patients and medical professionals did not suspect an RD, and (2) patients and medical professionals failed to access the accurate information when they did suspect an RD (Fig 1). Among these factors, we consider the first factor to be of particular importance. As described in the Results section, many of the participants had been living with symptoms for years without being aware of the possibility of RD and then one day they suddenly learned about HAE and were diagnosed. However, a small number of participants continued to seek a diagnosis during the undiagnosed period.

This finding (1) is consistent with the description of the EURODIS report. It pointed out that arriving at a correct diagnosis requires a crucial, but often difficult, step: a recognition that their disease is not one frequently encountered, but possibly an RD [8]. Also, based on our results, we reviewed other literature and found patient narratives from qualitative reports of Fabry's disease and epidermolysis bullosa that described similar experiences. During the undiagnosed period, some patients were not suspected with the possibility of RDs when they visited the hospital, and they gradually became accustomed to the treatment they received and stopped going to the hospital [10, 31, 32]. However, these reports did not describe the detailed reasons regarding why it is difficult to suspect an RD. To better understand what happens during the undiagnosed period, careful analysis of the physical and psychological states of patients is required; to the best of our knowledge, our study is one of the internationally significant efforts in this direction and has revealed the details of patients' experiences, struggles, and events during the undiagnosed period.

In Japan, the policy for medical research, such as the IRUD, has been launched for several years. IRUD claims that it has an established national network for rare and undiagnosed diseases diagnosis covering the entire geographic areas and specialty/subspecialty medical fields [33]. However, it would seem that this project and policy has focused on diagnosing patients who are "suspected" of having rare and intractable diseases. Thus, our study is significant in that it highlights the need to look at the stage that precedes "a suspicion of an RD."

## Why did HAE patients and their medical professionals not suspect an RD?

Why did neither the patients nor their medical professionals suspect an RD during the undiagnosed period? Our results show the following three obstacles: (1) medical professionals were not fully aware of the patient's overall condition, (2) insufficient knowledge about RDs and

lack of good partnership between medical professionals and patients, and (3) patients got used to the condition and stopped visiting the hospital. Below, we describe these obstacles in this order.

**Medical professionals were not fully aware of the patient's overall condition.** This study showed that the consultation behavior of patients with HAE differs depending on the site of symptom's onset. For visible swellings, patients questioned the cause and repeatedly visited hospitals (e.g., dermatology) until they eventually stopped their visits due to the belief that they would not get a clear answer. For abdominal symptoms, patients mainly expect treatments and continued visiting their usual hospitals depending on the extent of severity of the symptoms. This finding is consistent with a recent quantitative study of HAE which revealed that patients visited various department specialties depending on the site of swelling [27].

Considering this, we believe that there is a difference between the problems perceived by patients and the problems understood by medical professionals. Even if patients have questions about recurring physical problems, they may not think of the association of symptoms in different parts of the body. They may accept such symptoms as due to vague "physical weaknesses". Therefore, they visit hospitals mainly for problems in their daily lives. In medicine, diagnosis and treatment are made mostly based on symptoms in specific parts of the body and cared for by specialists in different departments. However, in the case of systemic RDs that develop symptoms in various parts of the body, the situation may be different. The diagnosis may be delayed if medical professionals do not take a systematic and comprehensive look at abnormalities with proper knowledge and suspicions of such diseases. This situation has been pointed out in previous articles about RDs; i.e., "seemingly unrelated symptoms" and "a lack of integrative provider education" are challenges that result in a diagnostic delay [34, 35].

This obstacle may be particularly noticeable in HAE. This is because HAE is characterized by large individual differences in symptoms, the possibility of symptoms occurring in multiple locations, and the fact that these symptoms overlap very closely with other common life symptoms.

**Insufficient knowledge about RDs and lack of good partnership between medical professionals and patients.** The results of this study also showed that patients often did not receive referrals to higher-order hospitals, even after several visits to the hospitals wondering about the cause of unexplained visible swelling or repeated visits to hospitals seeking treatment for abdominal symptoms.

One reason for these experiences may be a lack of knowledge among medical professionals, particularly in primary care. It has been demonstrated that there is a lack of awareness among physicians regarding HAE [36], and RDs in general [6, 37, 38].

Another reason is the problem of communication between medical professionals and patients [10]. For example, patients may have too much trust in the doctor's judgment to actively question the diagnosis, even if their symptoms do not improve. It is also known that misdiagnoses were associated with delays that were twice as long as those reported by patients who were correctly diagnosed at the first instance [8].

**Patients got used to the condition and stopped going to the hospital.** This study also clarified that some HAE participants had stopped visiting the hospital after repeated visits. Our study presented several reasons for this: the participants either thought their physical problem was a small matter or stress-related, or they lost hope in the hospital. These experiences have been observed for other RDs as well. Many RD patients undergo psychological and psychiatric explanations or treatments [8, 39, 40]. In addition, previous research showed that some patients without a diagnosis experienced mistrust for the medical profession [12]. These reasons prevented patients from seeking another diagnosis. The less patients seek out medical

consultations, the less of an opportunity there is for medical professionals to suspect an RD. This can also be considered as a cause of the prolonged undiagnosed period.

## How can we shorten the diagnostic delay?

The difficulty in suspecting RD found in HAE may be a problem in the case of other RDs as well. Therefore, one key solution should be to establish a system with various stakeholders to appropriately suspect RDs in general. In this section, we will propose two concrete measures that would be necessary to achieve this: (1) improving awareness of RDs, and (2) establishing an improved medical system for diagnosis.

**Improving awareness of RDs.** As mentioned above, one factor that prevents patients and their medical professionals from suspecting RDs is the lack of recognition. However, it is difficult to make a correct diagnosis for the 10,000 RDs in all of the hospitals. Therefore, it is necessary to educate all medical professionals about the existence rather than precise diagnosis of "rare diseases" or diseases that are difficult to diagnose [10, 35, 41, 42]. Especially, all clinicians should be trained to be able to realize the possibility of RDs.

The results of this study also indicate the importance of raising awareness of RDs among patients and the public. It is because medical professionals may not be fully aware of their patient's overall condition, and they may not be able to think of the possibility of RD. If patients understand these issues and act themselves, the possibility of suspecting an RD will increase. To raise awareness, other stakeholders such as mass media and pharmaceutical companies can play a key role.

Awareness of RDs will encourage patients and medical professionals to recall the suspicions regarding RDs at appropriate times. In this phase, collaboration between patients and medical professionals has been expected to contribute to the prevention of misdiagnoses [43]. If treatments are not effective, or if the cause of an abnormality is not explained at all, medical professionals must check if the patient has had such an experience before, or if the patient has any abnormalities in the body other than the symptom they are complaining of at the time [44]. Patients should also ask their doctors their question, for example, by using the checklist [35] to suspect if they have an RD.

**Establishing an improved medical system for diagnosis.** The improvement of medical systems is also an important issue. In this area, we expect to strengthen the coordination of the diagnostic processes. There are various ways to do this such as utilizing digital tools.

First, it is necessary to strengthen the coordination of diagnosis through policies and institution. Especially, coordination between local clinics and higher-order hospitals should be strengthened. It may be difficult for doctors at local clinics to determine which department to refer patients to even if they suspect the possibility of an RD. We need to establish centers that specialize in the diagnosis of RDs and consolidate patients suspected of having RDs.

Second, we can expect digital tools to facilitate the coordination process. For example, a questionnaire for differentiating patients with RDs from those of other disease groups, such as non-rare chronic or psychiatric diseases has been developed [45]. Also, AI tools for diagnostic support have already been developed to present possible RDs to patients who input their symptoms. These could be used to assist in arriving at a specific diagnosis [34, 46, 47].

## Methodological limitations

This study has methodological limitations. Participant bias existed in the following two aspects. First, most of the participants in this study were in their 40s to 60s. With HAE, it is known that the average year of undiagnosed period differs by generation [22], due to the increased awareness in recent decades. When considering the circumstances surrounding the

diagnosis of patients with HAE, the age at which the participants experienced the undiagnosed period is important. Therefore, this study may not capture the experiences of a younger generation of HAE patients. Second, many patients reported that their symptoms were worse than what is average for HAE patients. Among patients with relatively mild symptoms, the factors of prolonged undiagnosed period may differ from the findings of this study.

## Conclusions

In this study, we explored the experiences of patients with HAE who had remained undiagnosed for a long period of time. We found that during the undiagnosed period of HAE, only a small number of participants continued to seek a diagnosis. Many of the participants had been living with symptoms for years without being aware of the possibility of RD. This led to a long undiagnosed delay.

Concerning the activities to shorten the undiagnosed period, the current policy tends to focus on the period from suspecting RDs to a clear diagnosis. However, this study suggested that it is also necessary to focus on how to suspect RDs.

In future research, the experiences of other RD patients need to be explored. If we conduct an interview survey on more diseases with long undiagnosed periods, we may be able to elucidate common characteristics of the diseases with long delay of diagnosis. Furthermore, since factors related to the prolonged undiagnosed period are influenced by the social and cultural backgrounds as well as the medical system of the country of residence [48], it may be necessary to conduct a more detailed analysis of social factors regarding the reasons as to why patients are unable to suspect an RD.

## Supporting information

**S1 File. Interview guide.**
(PDF)

## Acknowledgments

We thank all the interviewees who have participated in this study. We also thank the patients with HAE and specialists who helped with the recruitment. We gratefully acknowledge Dr. Michihiro Hide and Ms. Kate Nakasato for giving valuable feedback for a draft of this manuscript. Dr. Madison from Editage Group and Mr. Joshua Wittig are acknowledged for language editing a draft of this manuscript. A part of the result has been orally presented at the 32th Annual meeting of the Japan Association for Bioethics in Japanese in December 2020.

## Author Contributions

**Conceptualization:** Moeko Isono, Kazuto Kato.

**Data curation:** Moeko Isono.

**Formal analysis:** Moeko Isono, Minori Kokado, Kazuto Kato.

**Funding acquisition:** Kazuto Kato.

**Investigation:** Moeko Isono, Minori Kokado, Kazuto Kato.

**Methodology:** Moeko Isono, Minori Kokado, Kazuto Kato.

**Project administration:** Moeko Isono, Kazuto Kato.

**Writing – original draft:** Moeko Isono.

**Writing – review & editing:** Moeko Isono, Minori Kokado, Kazuto Kato.

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
