## [Decision Letter · Decision Letter 0]

13 Dec 2021

PONE-D-21-28571Why does it take so long for rare disease patients to get an accurate diagnosis? - A qualitative investigation of patient experiences of hereditary angioedemaPLOS ONE

Dear Dr. Kato,

Thank you for submitting your manuscript to PLOS ONE. After careful consideration, we feel that it has merit but does not fully meet PLOS ONE’s publication criteria as it currently stands. Therefore, we invite you to submit a revised version of the manuscript that addresses the points raised during the review process.

Based on the reviewer's comments and careful evaluation, this manuscript need minor revision.

We look forward to receiving your revised manuscript.

Kind regards,

Girijesh Kumar Patel, PhD

Academic Editor

PLOS ONE

2. Pease provide the interview guide used as a Supplementary File.

Important: If there are ethical or legal restrictions to sharing your data publicly, please explain these restrictions in detail. Please see our guidelines for more information on what we consider unacceptable restrictions to publicly sharing data: http://journals.plos.org/plosone/s/data-availability#loc-unacceptable-data-access-restrictions. Note that it is not acceptable for the authors to be the sole named individuals responsible for ensuring data access

Reviewers' comments:

Reviewer's Responses to Questions

**Comments to the Author**

1. Is the manuscript technically sound, and do the data support the conclusions?

Reviewer #1: Yes

Reviewer #2: Yes

2. Has the statistical analysis been performed appropriately and rigorously? 

Reviewer #1: N/A

Reviewer #2: N/A

3. Have the authors made all data underlying the findings in their manuscript fully available?

Reviewer #1: Yes

Reviewer #2: Yes

4. Is the manuscript presented in an intelligible fashion and written in standard English?

Reviewer #1: Yes

Reviewer #2: Yes

5. Review Comments to the Author

Reviewer #1: Dear Authors,

First of all, I would like to congratulate and thank you for addressing the problem of such a complex and heterogeneous subject. The language of the manuscript is scientific, unambiguous, and explicit. The whole manuscript is highly engaging for me as a reviewer and an ordinary person from the masses. I am sure this information would add some new knowledge to the scientific field as well as this manuscript may guide some readers who may have HAE.

Here, there are some suggestions from my side:

1-Authors may add a table of the rare diseases which are hard to diagnose and and have no specific symptoms.

2-It would be great if the authors put a little bit of emphasis on diagnosing rare diseases in low and middle-income countries, too, as they face different challenges like socio-economic, limited resources, lack of data services, and competing for health priorities.

3-In this manuscript, a paragraph describes the recent and current projects and policies in different developed countries (from lines 66 to 77) to address this issue. Some developing countries where rare diseases are mishandled because of the diagnostic issues have started taking the initiative to address this problem. Ministry of Health and Family Welfare, Government of India formulated a National Policy for Treatment of Rare Diseases (NPTRD) in July 2017. A group of volunteers created a not-for-profit organization named Organization for Rare Diseases India (ORDI; www.ordindia.org). The North-West University's Centre for Human Metabolomics (CHM) establishes the first rare disease (RD) biobank in South Africa and Africa. The author may add information about some of these policies.

Specific suggestions about HAE:

1-There should be a little more description about the HAE, mainly different types of HAE (type I, II, and III), the possible genetic and molecular alteration and damage mechanism, and symptoms of each type.

2-Authors may add the information about the difference between acquired angioedema and HAE to clear the confusion between the two.

5-Authors may add some information if HAE is specific to any ethnicity or race.

6-The methodologies are excellent, but I found the sample size very small; I know it is a rare disease, so collecting many samples is challenging. My suggestion for the authors (for the future) is to collaborate with other countries like India, Iran, and South Africa, where many patients are diagnosed with HAE and information is available. I am not sure if it is possible or not.

6-Authors may add some of the clinical reports of the interviewees

7-Authors may add the information about specific blood tests to confirm Hereditary Angioedema Type I, II, or III.

General minor concerns:

1-There are some space issues, which may be formatted very quickly, for example, line 627.

2-The heading in line 542 needs to be corrected.

Reviewer #2: Working in the field of rare diseases is a challenge due to low sample number and delayed diagnosis. In this aspect, authors have made good and repeated effort in collecting/enrolling the hereditary angioedema rare disease patients in the survey. The overall findings of this qualitative study are interesting, especially the possible reasons for delay in the diagnosis of rare diseases. The approaches reported by patients in this study could be followed by other rare disease patients with challenges of diagnosis.

That being said, I would have been more favorable for this article if the authors included similar studies on at least another one rare disease to see the generality of findings of this study. I believe, even though this is only a qualitative study, the interpretations from patient survey in this study will provide important directions to other patients with rare diseases.

6. PLOS authors have the option to publish the peer review history of their article (what does this mean?). If published, this will include your full peer review and any attached files.

Reviewer #1: **Yes: **sabiha khatoon

Reviewer #2: **Yes: **Subash Kairamkonda

---

## [Author Response · Author response to Decision Letter 0]

6 Feb 2022

Dear Dr. Patel:

Thank you for inviting us to submit a revised draft of our manuscript. We also appreciate the time and effort you and the reviewers have dedicated to providing insightful feedback on ways to strengthen our paper.

We have responded to all the individual comments by the reviewers in a point-by-point fashion. Responses are indicated with blue colored font. In addition, we found two errors in the expressions in Table 4 and have corrected them. The article has undergone a round of proof-reading by a professional language editing service to improve the language, grammar, and overall tone of the manuscript. We apologize for the inconvenience caused as a result of us making corrections after the peer review.

We hope that the manuscript is positively received by the reviewers and editors.

Thank you very much for your consideration.

Sincerely,

Kazuto Kato

(Please note that the page and line numbers in our response correspond to those shown in the manuscript in our computer. They may slightly change depending on the computer environment.)

Reviewer #1

1. Authors may add a table of the rare diseases which are hard to diagnose and have no specific symptoms.

Thank you for your suggestion. It would have been interesting to explore this aspect, but, unfortunately, after considering the possibility of preparing a table of rare diseases that are hard to diagnose and have no specific symptoms, we reached a conclusion that it would not be possible to tabulate this information, due to two reasons. First, it is difficult to define ‘rare diseases which are hard to diagnose’. It is known that most rare diseases are difficult to diagnose. Even if we understand it as ‘rare diseases with long reported periods of not being diagnosed’, we are faced with the problem that there is no uniform definition of the undiagnosed days (i.e.,. the definition of start and end points, whether the time period is calculated using the mean or median). Second, we also think that it is difficult to select rare diseases that have no specific, identifying symptoms. Many rare diseases are said to share symptoms with ‘common’ diseases. In addition, each patient suffering from the same rare disease can have a multitude of different signs and symptoms. We believe it is important to clarify which diseases are difficult to diagnose so that more attention is given to them. Consequently, we hope we will be able to tackle these issues in the future work. 

2. It would be great if the authors put a little bit of emphasis on diagnosing rare diseases in low and middle-income countries, too, as they face different challenges like socio-economic, limited resources, lack of data services, and competing for health priorities.

Thank you for highlighting this important point. We have added this information about developing countries and regions in the revised manuscript (lines 57–59).

3. In this manuscript, a paragraph describes the recent and current projects and policies in different developed countries (from lines 66 to 77) to address this issue. Some developing countries where rare diseases are mishandled because of the diagnostic issues have started taking the initiative to address this problem. Ministry of Health and Family Welfare, Government of India formulated a National Policy for Treatment of Rare Diseases (NPTRD) in July 2017. A group of volunteers created a not-for-profit organization named Organization for Rare Diseases India (ORDI; www.ordindia.org). The North-West University's Centre for Human Metabolomics (CHM) establishes the first rare disease (RD) biobank in South Africa and Africa. The author may add information about some of these policies.

We appreciate your comment on this point. We have accordingly added the description to the revised manuscript (lines 68–75).

Specific suggestions about HAE

1. There should be a little more description about the HAE, mainly different types of HAE (type I, II, and III), the possible genetic and molecular alteration and damage mechanism, and symptoms of each type.

We thank you for this comment. We agree that this point requires additional clarification and have consequently added the information to the revised manuscript (lines 102–112, 115–116). However, we have kept the description of type 3 (i.e., HAE with normal C1-INH function) to a minimum. Type 3 is a very rare form of hereditary angioedema and is not diagnosed by C1-INH activity, so our study does not cover it.

2. Authors may add the information about the difference between acquired angioedema and HAE to clear the confusion between the two.

We appreciate your comment. We have accordingly added an explanation about acquired angioedema in the section describing HAE. (lines 110–112).

3. Authors may add some information if HAE is specific to any ethnicity or race.

Thank you for the valid query. We have added the relevant information to the revised manuscript (108–109). 

4. The methodologies are excellent, but I found the sample size very small; I know it is a rare disease, so collecting many samples is challenging. My suggestion for the authors (for the future) is to collaborate with other countries like India, Iran, and South Africa, where many patients are diagnosed with HAE and information is available. I am not sure if it is possible or not.

We agree that gathering additional information, as you suggested, would be valuable. It would be a very good idea to carry out collaborations with medical professionals and patients in those countries. For our present work, however, we do not think that it is possible to carry out these collaborations in a short time, particularly, because we do not have any connections with HAE specialists and patients with HAE in other countries. We would definitely like to consider this as a direction for future research.

5. Authors may add some of the clinical reports of the interviewees.

We appreciate your suggestion. We would like to point out is that in this study the interviewees were patients who self-reported their HAE, and we did not collect detailed clinical information. However, we have added some additional information about the severity of the symptoms experienced by each patient (Table 1). 

6. Authors may add the information about specific blood tests to confirm Hereditary Angioedema Type I, II, or III.

Thank you for the comment. We agree with you and recognize the importance of describing the information about the diagnostic tests for HAE. However, we believe it is important to point out that some of the diagnostic criteria set out in Japan do not necessarily correspond to those set out by the World Allergy Organization (WAO) in collaboration with the European Academy of Allergy and Clinical Immunology (EAACI). In other words, it is our humble opinion and belief that incorporating the precise and detailed information on this aspect is not feasible because it is not directly relevant to the main purpose and scope of this study. Therefore, we decided not to include the information about blood tests. We hope that you can understand our viewpoint.

General minor concerns

1. There are some space issues, which may be formatted very quickly, for example, line 627. 

Thank you for pointing it out. We have made the necessary change.

2. The heading in line 542 needs to be corrected.

This error has been corrected in accordance with your comment (line 565).

Reviewer #2

Working in the field of rare diseases is a challenge due to low sample number and delayed diagnosis. In this aspect, authors have made good and repeated effort in collecting/enrolling the hereditary angioedema rare disease patients in the survey. The overall findings of this qualitative study are interesting, especially the possible reasons for delay in the diagnosis of rare diseases. The approaches reported by patients in this study could be followed by other rare disease patients with challenges of diagnosis.

That being said, I would have been more favorable for this article if the authors included similar studies on at least another one rare disease to see the generality of findings of this study. I believe, even though this is only a qualitative study, the interpretations from patient survey in this study will provide important directions to other patients with rare diseases.

You have raised a crucial point. We agree that there is a need to demonstrate whether the results can be generalized to other diseases. As one of our future research works, we think that we should plan a research project on this issue, but unfortunately, for our current work, it would be difficult to conduct an additional research in a short time. We therefore reviewed the previous literature based on the results of our study. As a result, we found patient narratives from qualitative reports of Fabry's disease and epidermolysis bullosa that described similar experiences. Based on this, the description in the Discussion section has been modified (lines 453–457 of the revised manuscript). In a future study, we would like to investigate the generalizability of our findings to other diseases. Thank you for your valuable comment. 

There has been substantial modification to the paper in line with the reviewers’ suggestions, and I hope that the paper will benefit from these revisions. Once again, we thank you for the time you put in reviewing our paper and look forward to meeting your expectations.

---

## [Decision Letter · Decision Letter 1]

9 Mar 2022

Why does it take so long for rare disease patients to get an accurate diagnosis? - A qualitative investigation of patient experiences of hereditary angioedema

PONE-D-21-28571R1

Dear Dr. Kato,

We’re pleased to inform you that your manuscript has been judged scientifically suitable for publication and will be formally accepted for publication once it meets all outstanding technical requirements.

Kind regards,

Girijesh Kumar Patel, PhD

Academic Editor

PLOS ONE

Additional Editor Comments (optional):

Reviewers' comments:

Reviewer's Responses to Questions

**Comments to the Author**

1. If the authors have adequately addressed your comments raised in a previous round of review and you feel that this manuscript is now acceptable for publication, you may indicate that here to bypass the “Comments to the Author” section, enter your conflict of interest statement in the “Confidential to Editor” section, and submit your "Accept" recommendation.

Reviewer #1: All comments have been addressed

Reviewer #2: All comments have been addressed

2. Is the manuscript technically sound, and do the data support the conclusions?

Reviewer #1: Yes

Reviewer #2: Yes

3. Has the statistical analysis been performed appropriately and rigorously? 

Reviewer #1: Yes

Reviewer #2: N/A

4. Have the authors made all data underlying the findings in their manuscript fully available?

Reviewer #1: Yes

Reviewer #2: Yes

5. Is the manuscript presented in an intelligible fashion and written in standard English?

Reviewer #1: Yes

Reviewer #2: Yes

6. Review Comments to the Author

Reviewer #1: Dear Authors.

Thank you all for addressing all suggestions and concerns specified by me. I want to congratulate and thank you all for writing such an informative report after reading the current version of the manuscript. According to my view your manuscript is of very highly importance and ready to publish.

Reviewer #2: The authors have addressed all the comments and concerns of reviewers to satisfaction. The language edits have improved the manuscript's readability and made it easy to understand for the readers.

7. PLOS authors have the option to publish the peer review history of their article (what does this mean?). If published, this will include your full peer review and any attached files.

Reviewer #1: **Yes: **Sabiha Khatoon

Reviewer #2: **Yes: **Subash Kairamkonda

---

## [Editor Report · Acceptance letter]

11 Mar 2022

PONE-D-21-28571R1 

Why does it take so long for rare disease patients to get an accurate diagnosis? - A qualitative investigation of patient experiences of hereditary angioedema 

Dear Dr. Kato:

I'm pleased to inform you that your manuscript has been deemed suitable for publication in PLOS ONE. Congratulations! Your manuscript is now with our production department. 

Kind regards, 

on behalf of

Dr. Girijesh Kumar Patel 

Academic Editor

PLOS ONE